# RL-SSI Model: Adapting a Supervised Learning Approach to a Semi-Supervised Approach for Human Action Recognition

Lucas Lisboa dos Santos [1,†] , Ingrid Winkler [2,†] and Erick Giovani Sperandio Nascimento [1,*,†]

1 Department of Stricto Sensu Studies, SENAI CIMATEC University Center, Salvador 41650-010, Brazil; lucas.lisboa.santos@gmail.com
2 Department of Management and Industrial Technology, SENAI CIMATEC University Center, Salvador 41650-010, Brazil; ingrid.winkler@doc.senaicimatec.edu.br
* Correspondence: erick.sperandio@fieb.org.br
† These authors contributed equally to this work.

**Abstract:** Generally, the action recognition task requires a vast amount of labeled data, which represents a time-consuming human annotation effort. To mitigate the dependency on labeled data, this study proposes Semi-Supervised and Iterative Reinforcement Learning (RL-SSI), which adapts a supervised approach that uses 100% labeled data to a semi-supervised and iterative approach using reinforcement learning for human action recognition in videos. The JIGSAWS and Breakfast datasets were used to evaluate the RL-SSI model, because they are commonly used in the action segmentation task. The same applies to the performance metrics used in this work-F-Score (F1) and Edit Score-which are commonly applied for such tasks. In JIGSAWS tests, we observed that the RL-SSI outperformed previously developed state-of-the-art techniques in all quantitative measures, while using only 65% of the labeled data. When analysing the Breakfast tests, we compared the effectiveness of RL-SSI with the results of the self-supervised technique called SSTDA. We have found that RL-SSI outperformed SSTDA with an accuracy of 66.44% versus 65.8%, but RL-SSI was surpassed by the F1@10 segmentation measure, which presented an accuracy of 67.33% versus 69.3% for SSTDA. Despite this, our experiment only used 55.8% of the labeled data, while SSTDA used 65%. We conclude that our approach outperformed equivalent supervised learning methods and is comparable to SSTDA, when evaluated on multiple datasets of human action recognition, proving to be an important innovative method to successfully building solutions to reduce the amount of fully labeled data, leveraging the work of human specialists in the task of data labeling of videos, and their respectives frames, for human action recognition, thus reducing the required resources to accomplish it.

**Keywords:** deep learning; human action recognition; data annotation; semi-supervised learning; action segmentation

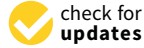



## 1. Introduction

The development of artificial intelligence techniques, more specifically Deep Learning (DL), allows several applications to be used as facilitators or as support for the solution of common everyday problems. Some of these problems may be related to the recognition of human actions, such as instructions to perform a task, or the detection of faulty movements in activities that require precision. In computer vision, the term "action recognition" is often treated as a pattern recognition problem with an additional time dimension to recognize human actions in videos [1].

The studies in the area of action recognition involve the analysis of images, understanding that each image is a frame under a temporal approach (the sequence of the frames, a determining factor for action recognition), which can portray different contexts (such as a person performing karate or fixing a car) and classes (in the karate context, each move

performed by the person is a class). There are also models capable of automatically learning the steps necessary to perform a task based on videos with narration and subtitles.

The action recognition task is part of the modernization and adoption of intelligent systems in various sectors of society, in the context of Industry 4.0. There are a lot of applications, such as surveillance [2,3], video retrieval [4,5], entertainment [6,7], human-robot interaction [8,9], and autonomous vehicles [10,11].

These applications require large datasets to validate their effectiveness. So, the development of several datasets, such as the COmprehensive INstructional Video Analysis (COIN) [12] dataset, consisting of 11,827 videos and 180 tasks related to everyday life organized in a hierarchical structure (the contexts and their respective classes) with descriptive annotation of actions and temporal boundaries. Another example is EPIC-KITCHENS [13], a dataset consisting of 100 h of first-person videos, which depict various activities in the context of cooking and are annotated based on the narrations of the participants.

The data present in these datasets can be classified into two groups: unlabeled and labeled. Unlabeled data consists of samples without the presence of data containing information (label) that defines its meaning. On the other hand, labeled data has this information that classifies the data [14]. For example, an image of a kitchen sink is an unlabeled data, but when we write data for this image with the information "sink", it becomes characterized as labeled. To construct labeled data, it is necessary to manually map these features, which makes this process expensive for a human activity [15].

On the other hand, manual video labeling needs precise frame-by-frame labeling of actions, which makes the process time-consuming and challenging [16]. Ref. [17] states that manual video labeling isn't feasible for large sets because it is expensive and requires a domain expert (e.g., to annotate a karate video, you need an expert who understands exactly what the moves are in order to correctly identify them in classes).

This restriction is even more significant for deep learning and artificial neural networks (ANN) techniques, since they are mostly supervised, and require a large amount of labeled data for training and subsequent image recognition [18]. As an example, to apply a technique for recognizing actions performed in car maintenance tasks, each frame in a set of videos must be labeled, identifying each maintenance context and class, substantially increasing the difficulty in preparing datasets of videos to develop a machine learning model for a specific task, since an extensive effort is required for annotation, considering the amount of frames per second in a video, and its duration. So, in this research we aim to provide an innovative approach that reduces the reliance on labeled data, commonly required in supervised learning approaches, to build machine learning models to recognize human actions in videos, based on the hypothesis of creating solutions with reduced need for fully labeled data, thus mitigating this problem and advancing in the current state-of-the-art.

In this context, this study proposes the Semi-Supervised and Iterative Reinforcement Learning (RL-SSI), which adapts a supervised approach that uses 100% labeled data as part of a semi-supervised and iterative approach using reinforcement learning for human action recognition in videos.

In this way, the main contributions to the literature of this research are:

- to provide an innovative approach that reduces the reliance on having a large amount of labeled data, commonly required in supervised learning approaches, to build machine learning models to recognize human actions in videos;
- to adapt a supervised learning approach to a semi-supervised and iterative learning, using the reward strength in reinforcement learning;
- to compare and assess the approach with equivalent learning methods with publicly available datasets of human action segmentation, aiming at having similar or even better performance, but with less annotated data.

This paper is organized into four parts. Section 2 presents related works, Section 3 outlines the methods used, Section 4 details our findings, and Section 5 presents our final considerations and recommendations for future investigations.

## 2. Basic Concepts

ANN are algorithms designed to model the structure or function of biological neural networks, simulated computationally by programming. In short, they are a processor that operates in parallel and a distributed manner through processing units that are neurons, which have the propensity to store experimental knowledge (learning) to be used in the desired application. During the learning process, there is an orderly modification of the synaptic weights of the network [19].

Thus, the goal of an ANN is to acquire learning from patterns for existing problems. That is, there must be some initial information so that the ANN can build and generate patterns from that data [19]. In this sense, there are four forms of learning: supervised, unsupervised, semi-supervised, and reinforcement learning, described briefly below:

- Supervised learning

To understand how supervised learning occurs, one can take as a reference the work of a teacher: the teacher has the knowledge of the environment, but the neural network does not. So, the teacher provides the knowledge so that the neural network can learn it and, in the end, applies a test to check if there was learning [19]. In this approach, the existence of the teacher is crucial, which in practice means that the input data must be properly labeled. This is because, during the training process, there will be an expected output and, if the output of the ANN does not match what is expected, the weights will be updated so that it can learn the data patterns.

- Unsupervised learning

In this approach there is no teacher, that is, the input data is not labeled and there is no expected output. As such, unsupervised learning entails an analysis of the structures present in the data, the goal of which may be to reduce redundancy or to group similar data [19]

- Semi-supervised learning

This approach resembles unsupervised learning because it looks for innate patterns in the data. However, this method relies on a certain amount of labeled and unlabeled data to complete its learning. In other words, it is an approach that uses concepts from both supervised and unsupervised approaches [20].

- Reinforcement learning

Reinforcement learning (RL) is a computational approach to decision-making for goal-defined problems [21], in which an artificial agent learns its choice policy from interactions with the environment. At a given time interval, the agent observes the state of the environment and selects an action, which, as far as it is concerned, affects the environment. The agent earns a numerical reward for each action and updates its policy to maximize future rewards. This sequential decision-making process is formalized as a MDP M: = (S, A, P, R, $\gamma$) [21], where:

- $S$ represents a finite set of states
- $A$ represents a finite set of actions
- $P: S \times A \times S \rightarrow [0, 1]$ denotes state transition probabilities
- $R: S \times A \rightarrow R$ denotes a reward function for each action performed in a given state
- $gamma \in [0, 1]$ is the discount factor that balances the immediate and long-term rewards.

The agent improves with experience to optimize its policy *pi: $S \times A \rightarrow [0, 1]$*, usually stochastic (random origin). The goal is to maximize the future reward with a cumulative discount from the current step to the end of the learning episode [22]. In some ways, this approach is similar to methodologies used in the process of animal training.

Table 1 displays the main characteristics and differences of each of the learning approaches.

**Table 1.** Summarization of learning approaches.

| | |
|---|---|
| Supervised | Learning is obtained based on the labeled data that is presented to the RNA. Requires fully labeled data. |
| Semi-supervised | Part of the learning is obtained from the labeled data and part from the innate analysis of the data. Requires partially labeled data. |
| Unsupervised | Learning is achieved by analyzing the structures present in the data, the goal of which may be to reduce redundancy or to group similar data. There is no need for labels. |
| Reinforcement Learning | This approach is focused on the agent's interaction with the environment. With each action of the agent in the environment, a feedback is generated, which can be negative or positive. The goal of RL is to model the reward, so that correct actions in the network have a positive feedback and incorrect actions have negative feedback. Requires a well-defined application context such as states, actions, policy, and the reward. |

## 3. Related Works

Action segmentation is a task linked to action recognition that aims to identify and specify where each action present in a video starts and ends [16]. The Temporal Convolutional Network (TCN) [23] is an action segmentation technique, which uses a hierarchy of temporal convolutions to perform granular action segmentation or detection. The TCN uses pooling and upsampling to efficiently capture long-range temporal patterns. Furthermore, ref. [23] proves that TCN can capture action compositions and segment durations.

The Self-Supervised Temporal Domain Adaptation (SSTDA) [16] has reformulated the task of action segmentation in view of cross-domain problems with domain discrepancy caused by spatio-temporal variations, i.e., when there is a difference in action performance because it is performed by multiple persons. The SSTDA contains two self-supervised auxiliary tasks (binary domain prediction and sequential domain predictions), which work cooperatively to align cross domains that have integrated spatial features with local and global temporal dynamics features. This study has performed two experiments: one with part of the labels for training (65% of the labels) to learn the adaptive features of each temporal domain, and another with 100% of the labels.

The work of [22] has related the task of segmenting surgical gestures (when a gesture starts and when it ends) and assigning a label to untrimmed videos (long videos that have several classes in a single video with some data category specifying them). The innovation of their approach is in sequential decision-making, where an intelligent agent is trained using reinforcement learning. In other words, the agent interacts with the environment (which, in this case, means the videos) and makes decisions (actions), which are related to the temporal aspect and its classification. These actions receive positive or negative feedback, according to the expected response, which characterizes it as a supervised technique. Ref. [22] further argues that this methodology is integrated with temporal consistency, which allows it to reduce excessive segmentation (when segmentation creates many insignificant boundaries), which is common in the gesture segmentation task. In short, the approach works in two steps: (i) use of TCN [23] to extract temporal features and (ii) sending of these features to a Reinforcement Learning Network (RLN) modeled so that positive and negative rewards are given according to expected output (ground-truth).

In this work, we present a technique that advances the state of the art by developing a model that adapts a supervised approach to a semi-supervised and iterative approach from a methodology applied to reinforcement learning. Among the techniques described above, the methodology presented by [22] is considered fruitful for this proposal, given the use of reinforcement learning and its potential of reward strength to measure the quality

of the proposed labels. In addition, the model of [22] is supervised, which provides us with an important basis of comparison to ascertain whether our proposed adaptation was successful in achieving similarity or was superior by using only part of the labels.

## 4. Materials and Methods

The RL-SSI was applied to the model of [22] to be able to run the ensemble iteratively with a reduced number of labels so that, at each completed iteration, valid labels are generated. The methodology applied to Reinforcement Learning (RL) allows labels with the highest degree of confidence, as measured through the reward function, to be moved to training in the next iteration. In this way, the supervised nature of the approach described by [22] is overcome, thus enabling a solution that requires less labeled data. To run the RL-SSI, the dataset had to be split into three parts:

- Training part: it contains the original labels from the dataset for the training step in the first iteration. In the following iterations it receives the best labels generated in the flexible data part.
- Flexible data part: it contains predicted data, and from this predicted data the best 25% will be moved to the training part in the next iteration. The criterion used to obtain the top 25% best labels is based on the strength of the RL reward.
- Testing part: it contains a fixed dataset on which the evaluation metrics will be measured.

The decision to move 25% of the generated labels per iteration, was an empirical choice based on experiments because since the experiment is run in iterations, it is important to move a portion of the best-generated labels to be able to iterate again in cycles.

Figure 1 illustrates the operation of the parts of RL-SSI over the course of iterations. As can be seen practically, Part 1 increases by absorbing the 1st quartile of Part 2 over the iterations, i.e., the 25% best labels generated and ranked by the reward function. As far as it is concerned, Part 3 will always be the same for evaluating metrics across all iterations.

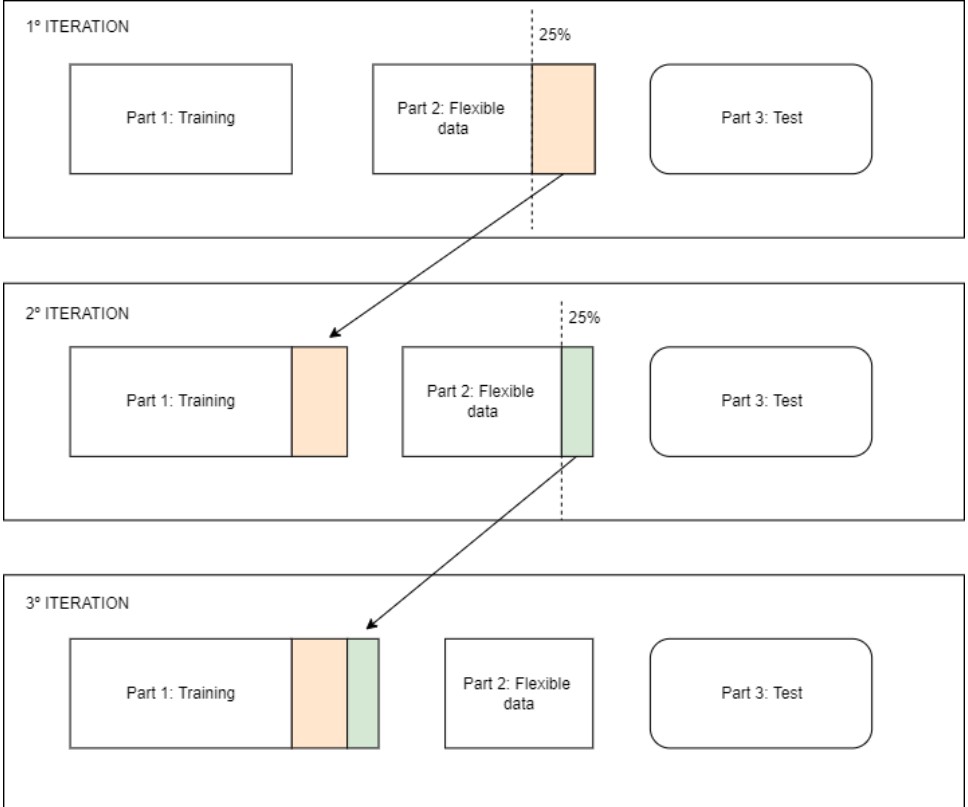

**Figure 1.** RL-SSI Data Structure.

In Figure 2 it is possible to visualize the standard execution flow of the RL-SSI applied to the model proposed by [22]. As can be seen, our proposal does not change or modify the TCN and RLN of the original model. However, our methodology RL-SSI exploits the reward strength of RL by using it as a criterion to select the top 25% of labels in each iteration.

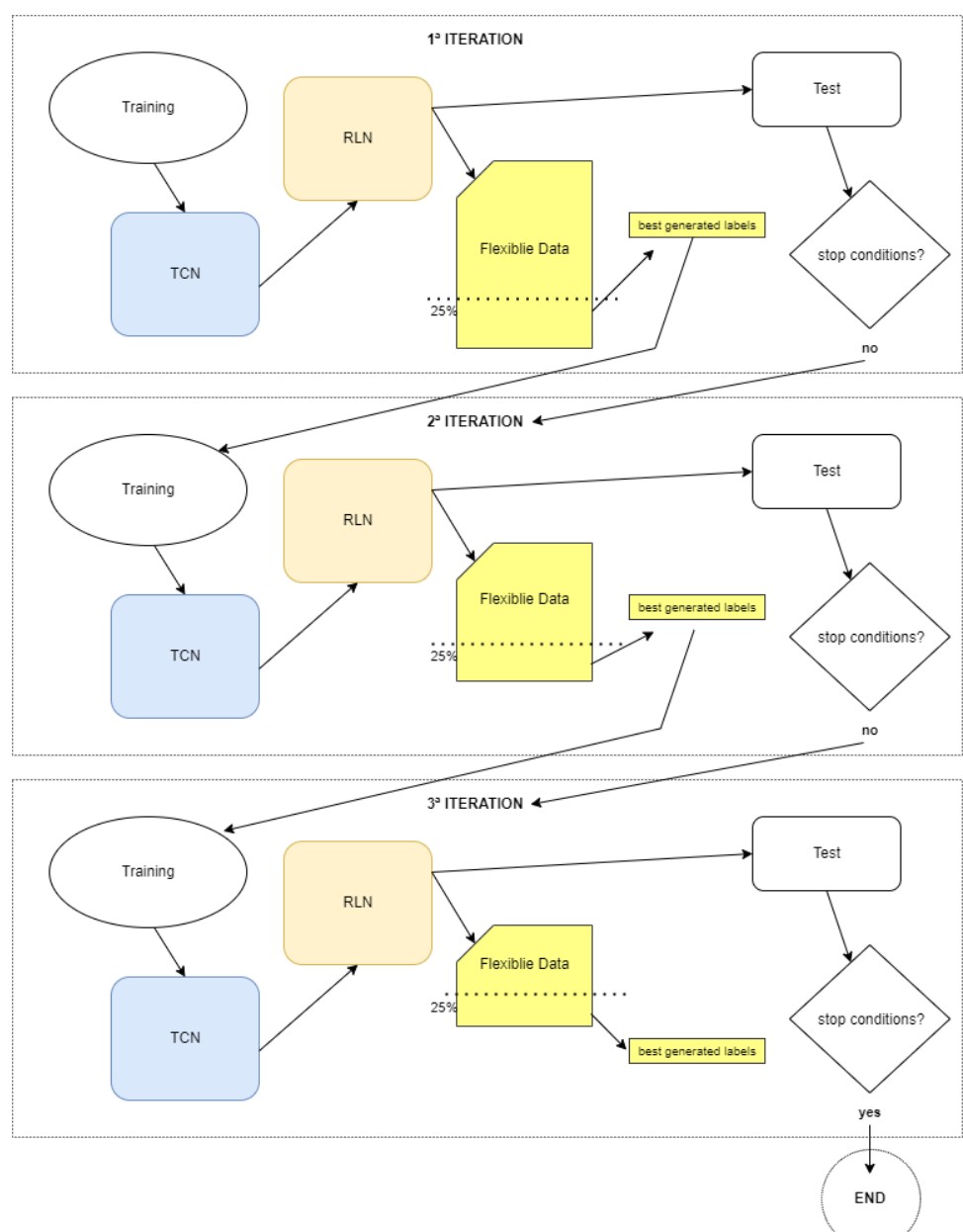

**Figure 2.** RL-SSI standard execution flow.

By establishing the three-part structure, we can see that the training part grows by absorbing the labels from the flexible data part over iterations, i.e., the 25% best labels generated and ranked by the reward function. As far as we are concerned, the testing part will always be the same to evaluate the metrics in all iterations. The standard RL-SSI execution flow does not alter or modify the TCN and RLN of the original model—that is, they function as independent blocks in the flow. However, the methodology proposed by the RL-SSI exploits the strength of reward of the RL by creating an iterative loop that changes the technique to a semi-supervised approach. As the experiment was designed to run cyclically and iteratively, a stopping condition needed to be developed to terminate the execution. This condition is reached when there is no improvement in up to three

consecutive runs, since the experiment loses the ability to present good results when subsequent iterations receive incorrect labels. In each interaction, three steps take place: (1) training of the TCN and RLN modules; (2) generation of valid labels; (3) a testing step where the iteration metrics are measured and the stopping condition is tested. When the stopping condition is not met in the testing step, the next iteration is started and the best labels from the generated flexible data are incorporated into the next training step.

### 4.1. Datasets

This research chose to use two datasets from the action segmentation task: the JIGSAWS and the Breakfast sets. The JIGSAWS is composed of visual data (videos), sensor data from the robotic hands (position, velocity, and angle), and manual annotation. It was collected using the Vinci Surgical System, operated by eight surgeons of different levels of experience load (hours of surgery already performed). For this reason, although the images depict robotic hands, it is possible to consider that the videos show human actions since they are reproducing and mimicking the movements performed by surgeons.

The footage features three elementary surgical activities which are Suturing, Needle Passing, and Knot Tying and the dataset has a total of 15 classes not necessarily present in the three activities [24]. There are a total of 103 videos with about 2 min long in average, totaling about three hours of recording. This set was used in the work of [22], making it the primary target of this research, since the proposed adaptation, to be described in the following sections, will be applied to it. The model presented in the work by [22] used only the Suturing context, thus the experiment developed by this research will use the same context to maintain the same comparison basis.

Breakfast is the largest dataset of human action segmentation tasks, with 77 h of recordings, more than four million frames and 10 contexts (Coffee, Milk, Juice, Tea, Cereals, Fried Eggs, Pancakes, Salad, Sandwich, and Scrambled Egg), totaling 48 classes not necessarily present in all contexts. The videos were recorded by 52 people in 18 kitchens, shot by three to five cameras. To reduce the size of the data, the videos were resized to $320 \times 240$ pixels with 15 FPS [25].

Because it is the largest dataset for human action segmentation tasks, it makes it possible to perform a large amount of experiments. Moreover, it is used by other researchers on the same task (segmentation), providing this research with a rich base of comparison of results. For these reasons this set was selected for the experiments.

### 4.2. Experiment Settings

To ascertain whether the proposed adaptation of the supervised model to a semi-supervised and iterative technique does not negatively impact the results, experiments were developed with two datasets—the JHU-ISI Gesture and Skill Assessment Working Set (JIGSAWS) [24] and the Breakfast [25]. Furthermore, the purpose of using two datasets is to gauge whether the RL-SSI performs well with data from different application areas (in this case, surgical gestures and breakfast preparation). Our interest in the JIGSAWS stems from the fact that it is the dataset used in [22], so it is possible to directly compare the gains and losses of applying the RL-SSI methodology to the supervised technique by making it semi-supervised and iterative. It is important to point out that only the Suturing context was used, totaling 39 videos. On the other hand, Breakfast was chosen for being the largest dataset mapped to the action segmentation task, which guarantees numerous experiments, besides being used by several references in the state of the art, which allows us to have concrete bases to compare the results of the RL-SSI. Thus, experiments were performed with all the contexts in the dataset, that is, 1989 videos. The experiments performed with Breakfast used, on average, 55.8% of the original labels reserved for training, while the JIGSAWS experiments used 65%. The execution of the experiments was performed individually by each context of the Breakfast and the Suturing of the JIGSAWS, for a total of 11 experiments. Each experiment was run with an NVIDIA Tesla P100- SXM2 video card

and the average execution time for each experiment (training, inferences in all iterations) was approximately three days.

### 4.3. Implementation

"For the implementation of the RL-SSI, some hyperparameters were established. The target metric used to test the stopping conditions of the experiment was F-Score (F1), with a threshold of 10%, reached when there are three consecutive lower values in the iterations. The work of [26] has introduced this metric to the stock segmentation task because it is efficient in classification and segmentation tasks with three important features: it penalizes excessive segmentation errors; it does not penalize small temporal changes between predictions and expected responses, given the variability that can occur in processes with different human annotators (for example, one annotator may consider that the beginning of the action takes place in frame 50, while another annotator may consider that this same beginning occurs in frame 53); and scoring depends more on the number of actions, not their size. Furthermore, ref. [26] states that the F1 values of the segments are better represented qualitatively.

A significant element for this choice is that it is the most relevant metric in segmentation challenges, such as the Davis Challenge [27], which uses it to rank the best works in the domain of multiple object segmentation in videos. Since the RL-SSI exploits a methodology applied to RL of the model itself without changing it internally, the parameters originally used in [22] have been kept. The TCN has 300 epochs, a learning rate of 0.00001, a batch size of 1, and a weight decay of 0.0001. The RLN, on the other hand, was implemented with the Python programming language and the OpenAI Baselines library [28]. The policy network Trust Region Policy Optimization (TRPO) is of a hidden layer with 64 hidden units, and the time steps of TRPO were set to a value of 500,000. The discount factor gamma and the reward weight alpha were kept as 0.9 and 0.1, respectively.

Another important parameter of [22] is the step type, which can be short ($ks$) or long ($kl$). The RLN will classify a number of frames depending on the steps set. This binary strategy enables the agent to change the step size based on confidence in the label of the action to be made. The agent can adopt the smaller step when the state is not discriminative enough, or the larger step otherwise. In each action of the agent, $k$ frames are labeled with the same class, explicitly enforcing temporal consistency.

However, the values of the $ks$ and the $kl$ needed to be recalculated, since they were defined as the size of the smallest action in the context and the shortest average action length for each class, respectively. The values calculated for each context of a dataset can be seen in Table 2.

**Table 2.** RLN Steps Values.

| Datasets | Contexts | Short Step | Long Step |
|---|---|---|---|
| Breakfast | Coffee | 2 | 81 |
| | Milk | 29 | 128 |
| | Juice | 25 | 80 |
| | Tea | 33 | 104 |
| | Cereals | 20 | 80 |
| | Fried Eggs | 16 | 83 |
| | Pancakes | 10 | 82 |
| | Salad | 8 | 110 |
| | Sandwich | 30 | 94 |
| | Scrambled Egg | 19 | 88 |
| JIGSAWS | Suturing | 4 | 21 |

## 5. Results and Discussion

Table 3 highlights the best results of the RL-SSI metrics for the experiment with JIGSAWS (the best results are in green). Since the model of [22] has two modules, one from TCN and one from RLN, we can benchmark the results produced in both. We can observe that RLN generated good results, although TCN performed even better for both accuracy and segmentation metrics. This highlights that, in the proposed adaptation, TCN has better benefits from the iterative semi-supervised process on this dataset. Even with this performance for TCN, RLN is critical in the proposed adaptation given that the applied methodology exploiting reward strength is crucial for the iterative process to be semi-supervised, and in this model under study this occurs after the action of the TCN.

Table 4 compares the model of [22] with the RL-SSI (the best results are in green). We can observe that RL-SSI obtained better results in the metrics Edit Score and F1@10,25,50, losing only in Accuracy to the results of the reference technique [22]. However, similarly to our prior findings, the best results of the iterations occurred in TCN and, in this case, RL-SSI outperformed in all quantitative metrics the model of [22]. Both RL-SSI and the model of [22] used the Suturing context of JIGSAWS, but [22] is a supervised approach using 100% of the labels, while RL-SSI used only 65% of them. Therefore, we conclude that the RL-SSI accomplishes its goal, that is successfully adapting a supervised technique to a semi-supervised and iterative technique, and it outperforms the results of the base technique on all quantitative metrics in the TCN results and on three of the four RLN metrics.

Tables 5–14 summarize the results of the RL-SSI in each Breakfast context (the best results are in green). We can observe that, unlike the JIGSAWS experiment, RLN was predominantly better than TCN in relation to the accuracy metrics and always better in relation to the segmentation metrics. In some cases such as in the context of Coffee, Tea, and Fried Egg the accuracy of TCN was slightly better as the differences were no more than two percentage points.

The best values varied between the first and third iteration, while the maximum amount of iterations of the experiments varied from four to six. Another notable pattern is that the best experiment was always the third, counting inversely from the last iteration. The Juice context obtained the best accuracy results and the Cereals context obtained the best results in relation to the segmentation metrics. The standard deviation for the accuracy metric was 6.37, which is lower than that of the Edit Score and F1@10,25,50 segmentation metrics, which were 14.28 and {14.21, 15.69, 15.05}, respectively. This highlights a greater variation with the segmentation metrics with respect to accuracy. Table 15 presents the result of each context with its best iteration, maximum amount of iterations, the arithmetic mean, and the standard deviation of all contexts to consolidate the comparative results of RL-SSI in the experiments with the Breakfast dataset.

Table 16 presents the comparison of the results of SSTDA [16] and RL-SSI (the best results are in green). As a result, RL-SSI outperformed SSTDA by obtaining an accuracy of 66.44% versus 65.8% and F1@50 of 63.19% versus 62.9%. However, RL-SSI was outperformed by SSTDA in the Edit metrics by obtaining 60.36% versus 69.0%, F1@10 of 67.33% versus 69.3%, and F1@50 of 47.86% versus 49.4%. When visualizing the standard deviation, it is possible to see that there was little variation in all metrics when values are below one, except for the Edit Score that obtained 4.32.

**Table 3.** RL-SSI in the JIGSAWS Dataset.

| | **TCN** | | | | | **RLN** | | | | |
|---|---|---|---|---|---|---|---|---|---|---|
| **Iteration** | **Acc** | **Edit** | **F1@{10,25,50}** | | | **Acc** | **Edit** | **F1@{10,25,50}** | | |
| 1 | 86.13 | 99.0 | 97.32 | 96.27 | 89.42 | 76.56 | 92.67 | 93.34 | 90.63 | 84.46 |
| 2 | 87.59 | 98.13 | 97.99 | 94.72 | 92.77 | 76.78 | 92.73 | 91.87 | 90.86 | 85.82 |
| 3 | 87.57 | 99.0 | 99.49 | 95.40 | 87.68 | 77.53 | 92.15 | 92.75 | 91.01 | 85.73 |
| 4 | 88.09 | 99.0 | 98.43 | 95.40 | 91.34 | 77.51 | 92.48 | 93.29 | 90.61 | 85.82 |

**Table 4.** Comparison of [22] and RL-SSI.

| | TCN | | | | | RLN | | | | |
|---|---|---|---|---|---|---|---|---|---|---|
| | **Acc** | **Edit** | **F1@{10,25,50}** | | | **Acc** | **Edit** | **F1@{10,25,50}** | | |
| Liu e Jiang | 81.71 | 86.63 | 91.0 | 89.5 | 82.0 | 81.43 | 87.96 | 92.0 | 90.5 | 82.2 |
| RL-SSI | 87.57 | 99.0 | 99.49 | 95.40 | 87.68 | 77.53 | 92.15 | 92.75 | 91.01 | 85.73 |

**Table 5.** RL-SSI in the Coffee context.

| | TCN | | | | | RLN | | | | |
|---|---|---|---|---|---|---|---|---|---|---|
| **Iteration** | **Acc** | **Edit** | **F1@{10,25,50}** | | | **Acc** | **Edit** | **F1@{10,25,50}** | | |
| 1 | 66.07 | 64.03 | 70.92 | 68.26 | 58.07 | 64.19 | 68.78 | 75.83 | 74.21 | 59.16 |
| 2 | 63.22 | 57.86 | 66.51 | 64.58 | 52.44 | 63.34 | 63.12 | 70.49 | 68.83 | 55.18 |
| 3 | 64.13 | 56.41 | 64.29 | 59.94 | 51.03 | 64.15 | 66.35 | 72.04 | 69.78 | 57.56 |
| 4 | 64.78 | 47.52 | 56.49 | 55.59 | 49.19 | 63.28 | 59.22 | 66.61 | 65.98 | 57.32 |

**Table 6.** RL-SSI in the Juice context.

| | TCN | | | | | RLN | | | | |
|---|---|---|---|---|---|---|---|---|---|---|
| **Iteration** | **Acc** | **Edit** | **F1@{10,25,50}** | | | **Acc** | **Edit** | **F1@{10,25,50}** | | |
| 1 | 78.35 | 48.54 | 59.33 | 56.88 | 48.48 | 78.85 | 67.38 | 73.91 | 72.52 | 60.78 |
| 2 | 76.7 | 49.5 | 58.9 | 56.59 | 48.37 | 76.94 | 64.77 | 72.79 | 71.54 | 58.76 |
| 3 | 77.94 | 46.4 | 56.03 | 52.53 | 43.37 | 78.06 | 62.31 | 71.08 | 68.74 | 56.34 |
| 4 | 79.42 | 46.38 | 56.23 | 54.85 | 46.57 | 79.33 | 66.28 | 73.41 | 72.37 | 63.75 |

**Table 7.** RL-SSI in the Milk context.

| | TCN | | | | | RLN | | | | |
|---|---|---|---|---|---|---|---|---|---|---|
| **Iteration** | **Acc** | **Edit** | **F1@{10,25,50}** | | | **Acc** | **Edit** | **F1@{10,25,50}** | | |
| 1 | 63.24 | 37.56 | 49.2 | 43.66 | 34.79 | 62.93 | 65.15 | 73.03 | 68.68 | 54.28 |
| 2 | 63.39 | 32.6 | 44.61 | 40.12 | 29.48 | 62.08 | 67.36 | 75.51 | 69.76 | 50.04 |
| 3 | 63.89 | 35.18 | 47.05 | 40.38 | 30.31 | 61.73 | 65.81 | 73.67 | 68.68 | 49.52 |
| 4 | 59.81 | 33.97 | 44.53 | 40.25 | 27.78 | 61.52 | 65.49 | 73.2 | 68.51 | 49.55 |
| 5 | 64.48 | 26 | 38.1 | 35.2 | 21.66 | 65.39 | 64.64 | 74.8 | 70.45 | 55.12 |

**Table 8.** RL-SSI in the Tea context.

| | TCN | | | | | RLN | | | | |
|---|---|---|---|---|---|---|---|---|---|---|
| **Iteration** | **Acc** | **Edit** | **F1@{10,25,50}** | | | **Acc** | **Edit** | **F1@{10,25,50}** | | |
| 1 | 64.7 | 50.05 | 60.02 | 57.67 | 50.79 | 62.6 | 67.26 | 74.43 | 71.27 | 57.11 |
| 2 | 63.54 | 49.28 | 57.75 | 54.57 | 39.95 | 61.62 | 67.17 | 71.97 | 66.16 | 53.77 |
| 3 | 65.7 | 49.09 | 57.3 | 53.69 | 43.72 | 64.91 | 71.52 | 77.79 | 74.56 | 58.15 |
| 4 | 65.14 | 44.16 | 52.45 | 49.29 | 40.33 | 64.67 | 67 | 74.38 | 70.6 | 58.76 |
| 5 | 62.97 | 43.95 | 53.56 | 50.28 | 40.33 | 61.07 | 65.31 | 71.34 | 68.62 | 51.49 |
| 6 | 65.09 | 48.48 | 58.58 | 56.68 | 43.14 | 62.86 | 66.77 | 74.43 | 70.43 | 56.15 |

**Table 9.** RL-SSI in the Cereals context.

| | TCN | | | | | RLN | | | | |
|---|---|---|---|---|---|---|---|---|---|---|
| **Iteration** | **Acc** | **Edit** | **F1@{10,25,50}** | | | **Acc** | **Edit** | **F1@{10,25,50}** | | |
| 1 | 69.8 | 68.1 | 73.3 | 70.30 | 58.0 | 69.5 | 75.6 | 80.6 | 76.8 | 62.4 |
| 2 | 69.0 | 54.6 | 63.5 | 58.35 | 43.6 | 69.8 | 76.1 | 82.5 | 78.7 | 61.5 |
| 3 | 67.4 | 45.3 | 54.1 | 50.56 | 38.7 | 68.5 | 73.1 | 79.8 | 76.2 | 59.8 |
| 4 | 67.3 | 45.7 | 54.7 | 51.38 | 36.7 | 68.3 | 75.3 | 80.8 | 76.8 | 57.5 |
| 5 | 66.7 | 44.4 | 53.6 | 49.64 | 37.4 | 67.6 | 73.9 | 79.4 | 74.8 | 57.7 |

**Table 10.** RL-SSI in the Fried Egg context.

| | | TCN | | | | | | RLN | | | |
|---|---|---|---|---|---|---|---|---|---|---|---|
| **Iteration** | **Acc** | **Edit** | **F1@{10,25,50}** | | | **Acc** | **Edit** | **F1@{10,25,50}** | | |
| 1 | 71.38 | 41.45 | 51.58 | 47.09 | 36.34 | 70.8 | 50.19 | 58.91 | 54.33 | 40.11 |
| 2 | 68.95 | 37.43 | 46.13 | 39.54 | 25 | 68.52 | 51.12 | 58.47 | 50.63 | 36.94 |
| 3 | 68.16 | 30.36 | 39.84 | 33.86 | 25.84 | 66.79 | 45.01 | 54.72 | 48.82 | 35.06 |
| 4 | 69.73 | 31.56 | 41.1 | 35.46 | 24.84 | 69.81 | 47.4 | 56.45 | 50.91 | 36.72 |

**Table 11.** RL-SSI in the Pancake context.

| | | TCN | | | | | | RLN | | | |
|---|---|---|---|---|---|---|---|---|---|---|---|
| **Iteration** | **Acc** | **Edit** | **F1@{10,25,50}** | | | **Acc** | **Edit** | **F1@{10,25,50}** | | |
| 1 | 62.09 | 31.4 | 41.13 | 37.23 | 24.57 | 61.29 | 37.99 | 47.51 | 43.44 | 30.16 |
| 2 | 63.67 | 29.21 | 36.04 | 30.59 | 20.59 | 63.62 | 36.67 | 43.83 | 38.89 | 26.24 |
| 3 | 63.65 | 28.27 | 37.04 | 32.52 | 24 | 64.38 | 35.56 | 43.63 | 40.39 | 29.55 |
| 4 | 61.13 | 24.87 | 33.28 | 30.3 | 19.83 | 61.88 | 34.24 | 43.3 | 39.03 | 27.29 |

**Table 12.** RL-SSI in the Salad context.

| | | TCN | | | | | | RLN | | | |
|---|---|---|---|---|---|---|---|---|---|---|---|
| **Iteration** | **Acc** | **Edit** | **F1@{10,25,50}** | | | **Acc** | **Edit** | **F1@{10,25,50}** | | |
| 1 | 58.95 | 35.04 | 38.52 | 29.06 | 17.44 | 59.61 | 43.56 | 46.64 | 38.02 | 22.94 |
| 2 | 61.88 | 37.88 | 39.06 | 31.5 | 18.47 | 62.05 | 44.83 | 44.84 | 35.87 | 22.92 |
| 3 | 61.59 | 32.38 | 35.89 | 31.3 | 17.44 | 61.9 | 43.85 | 46.37 | 40.32 | 23.45 |
| 4 | 61.76 | 32.32 | 36.17 | 30.08 | 16.82 | 62.86 | 41.89 | 44.36 | 37.92 | 22.95 |

**Table 13.** RL-SSI in the Sandwich context.

| | | TCN | | | | | | RLN | | | |
|---|---|---|---|---|---|---|---|---|---|---|---|
| **Iteration** | **Acc** | **Edit** | **F1@{10,25,50}** | | | **Acc** | **Edit** | **F1@{10,25,50}** | | |
| 1 | 71.68 | 40.52 | 52.37 | 49.76 | 40.67 | 72.26 | 53.95 | 65.73 | 63.06 | 50.23 |
| 2 | 70.21 | 27.14 | 39.38 | 37.58 | 26.05 | 73.93 | 52.68 | 64.91 | 62.7 | 53 |
| 3 | 73.4 | 39.47 | 49.52 | 48.89 | 39.75 | 73.52 | 59.18 | 68.84 | 66.71 | 54.79 |
| 4 | 73.65 | 31.37 | 43.15 | 39.74 | 31.83 | 74.19 | 55.96 | 66.56 | 62.39 | 54.47 |
| 5 | 74.08 | 30.06 | 42.45 | 38.29 | 27.41 | 75.08 | 53.38 | 65.12 | 62.84 | 50.19 |
| 6 | 74.41 | 34.56 | 44.67 | 43.56 | 32.42 | 75.25 | 53.1 | 64.15 | 61.57 | 48.97 |

**Table 14.** RL-SSI in the Scrambled Egg context.

| | | TCN | | | | | | RLN | | | |
|---|---|---|---|---|---|---|---|---|---|---|---|
| **Iteration** | **Acc** | **Edit** | **F1@{10,25,50}** | | | **Acc** | **Edit** | **F1@{10,25,50}** | | |
| 1 | 56.31 | 30.56 | 38.32 | 33.94 | 24.46 | 56.25 | 42.63 | 49.19 | 42.67 | 32.16 |
| 2 | 57.55 | 28.25 | 35.75 | 31.41 | 21.73 | 57.17 | 39.68 | 47.57 | 42.85 | 31.66 |
| 3 | 53.83 | 24.85 | 31.34 | 24.74 | 17.13 | 53.49 | 37.66 | 43.43 | 36.23 | 26.91 |
| 4 | 57.67 | 29.13 | 35.85 | 30.57 | 20.6 | 57.8 | 37.64 | 45.68 | 39.23 | 26.34 |

**Table 15.** RL-SSI in all breakfast contexts.

| Contextos | Best Iteration | Max. Iteration | Accuracy | Edit Score | F1@{10,25,50} | | |
|-----------|----------------|----------------|----------|------------|------|------|------|
| Coffe | 1st | 4 | 64.19 | 68.78 | 75.83 | 74.21 | 59.16 |
| Juice | 1st | 4 | 78.85 | 67.38 | 73.91 | 72.52 | 60.78 |
| Milk | 2nd | 5 | 62.08 | 67.36 | 75.51 | 69.76 | 50.04 |
| Tea | 3rd | 6 | 64.91 | 71.52 | 77.79 | 74.56 | 58.15 |
| Cereals | 2nd | 5 | 69.8 | 76.1 | 82.5 | 78.7 | 61.5 |
| Fried Egg | 1st | 4 | 70.8 | 50.19 | 58.91 | 54.33 | 40.11 |
| Pancakes | 1st | 4 | 61.29 | 37.99 | 47.51 | 43.44 | 30.16 |
| Salat | 1st | 4 | 59.61 | 43.56 | 46.64 | 38.02 | 22.94 |
| Sandwich | 3rd | 6 | 73.52 | 59.18 | 68.84 | 66.71 | 54.79 |
| Scrambled Egg | 1st | 4 | 56.25 | 42.63 | 49.19 | 42.67 | 32.16 |
| Arith. Mean | - | - | 66.44 | 60.36 | 67.33 | 63.19 | 47.86 |
| Standard Deviation | - | - | 6.37 | 14.28 | 14.21 | 15.69 | 15.05 |

**Table 16.** Comparison of SSTDA [16] and RL-SSI.

| | Acc | Edit | F1@{10,25,50} | | |
|---|-----|------|------|------|------|
| SSTDA | 65.8 | 69.0 | 69.3 | 62.9 | 49.4 |
| RL-SSI | 66.44 | 60.36 | 67.33 | 63.19 | 47.86 |
| Standard Deviation | 0.32 | 4.32 | 0.99 | 0.15 | 0.77 |

Thus, we can observe that RL-SSI outperformed SSTDA with a difference of 0.76% in accuracy, and with 0.29% for the F1@25 metric. However, it lost performance in the quality of segmentation with the Edit Score metric by 8.64%, by a difference of 2.03% for the F1@10 metric, and, for the F1@50 metric, by 1.54%. However, RL-SSI used only 55.8% of the labeled data, while SSTDA used 65% of them.

The experiments with the Breakfast dataset revealed that the application of the RL-SSI model applied in the transformation of the supervised approach, based on [22], into a semi-supervised and iterative one, was also successful in different contexts of human action recognition, in this way expanding the application of the proposed method. Moreover, the comparison with the SSTDA technique shows that RL-SSI presents competitive results using a reduced number of labels. This work advances the state-of-the-art by obtaining results that are competitive with fully supervised learning techniques, presenting the important characteristic of creating a solution that does not need to use a large number of labels.

## 6. Conclusions

The experiments with JIGSAWS have revealed that RL-SSI, which has successfully adapted a supervised technique to a semi-supervised and iterative technique, outperformed in all quantitative metrics the technique of [22] using only 65% of the annotated labels. On the other hand, the results of the Breakfast experiments show that RL-SSI is also successful for other action contexts, thus expanding the proposal to other applications. The RL-SSI proved to be competitive by overcoming the SSTDA with a difference of 0.76% in accuracy and 0.29% in F1@25. On the other hand, it underperformed in the segmentation quality with the Edit Score metric by 8.64%, the F1@10 metric by a difference of 2.03%, and F1@50 by 1.54%. However, it is worth noting that RL-SSI used only 55.8% of the labels, while SSTDA used 65%.

To mitigate the dependence on big amount of labeled data for building machine learning models for human action recognition, which is an existing gap in the literature, the experimental results of RL-SSI demonstrate that our approach outperformed equivalent supervised learning methods and is comparable to SSTDA when evaluated on multiple datasets, having an important innovative aspect by proving to be successful in its purpose of building solutions to reduce the need for fully labeled data, leveraging the work of

human specialists in the task of data labeling, thus reducing the required resources to accomplish it.

In future research, we envision application of the RL-SSI methodology in other supervised techniques, keeping its basic structure: execution in iterations, division of the data into three pieces (training, flexible data, and testing), and finally, some mechanism to measure and move the best-generated labels. As well as the possibility of making changes to the policy network aimed at optimizing the reward strength in the RLN. Finally, another recommendation would be to investigate the effect percentages of labeled data, such as 20%, 40%, 60%, and 80% of the original labeled data.

**Author Contributions:** Conceptualization, E.G.S.N. and L.L.d.S.; methodology, I.W.; software, L.L.d.S.; validation, E.G.S.N., I.W.; formal analysis, E.G.S.N. and I.W.; investigation, L.L.d.S.; resources, L.L.d.S.; data curation, L.L.d.S.; writing—original draft preparation, L.L.d.S.; writing—review and editing, E.G.S.N. and I.W.; visualization, L.L.d.S.; supervision, E.G.S.N. and I.W.; project administration, E.G.S.N. and I.W.; funding acquisition, E.G.S.N. and I.W. All authors have read and agreed to the published version of the manuscript.

**Funding:** Research funded by HP Inc. Brazil to be entitled to the financial credit defined in the Art. 4° of Law number 8.248, by 1991 (Computer Law).

**Institutional Review Board Statement:** Not applicable.

**Informed Consent Statement:** Not applicable.

**Data Availability Statement:** Not applicable.

**Acknowledgments:** We grateful acknowledge the support of SENAI CIMATEC AI Reference Center for the scientific and technical support and the SENAI CIMATEC Supercomputing Center for Industrial Innovation. The authors would like to thank for financial support from the National Council for Scientific and Technological Development (CNPq). Ingrid Winkler is a CNPq technological development fellow (Proc. 308783/2020-4).

**Conflicts of Interest:** There are no conflicts of interest associated with this publication.

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
