# Peer review of "RL-SSI Model: Adapting a Supervised Learning Approach to a Semi-Supervised Approach for Human Action Recognition"

_electronics, doi:10.3390/electronics11091471_

Round 1

Reviewer 1 Report

Overall, the paper is well written and contribute to the literature. My specific comments are as follows:

In the abstract, it would be important to explain the main contribution of the paper and indicate the sample and the period.

In the discussion, it would be important to explain the gap in the literature and the contribution of the paper to the literature. It is not clear how the paper contributed to the literature.

In the introduction it would be good to explain the research problem, research hypothesis and contribution of the paper.

The literature review is too sparse. It seems that some earlier work has been cited totally arbitrarily without following a logical plan that could motivate the paper. Please add some of the following references about intelligent algorithms:

Nebojša Denić, Dalibor Petković, Boban Spasić, Global Economy Increasing by Enterprise Resource Planning, Editor(s): Saleem Hashmi, Imtiaz Ahmed Choudhury,. Encyclopedia of Renewable and Sustainable Materials, Elsevier, 2020, Pages 331-337, ISBN 9780128131961, https://doi.org/10.1016/B978-0-12-803581-8.11590-5. (https://www.sciencedirect.com/science/article/pii/B9780128035818115905) 

Boban Spasić, Boris Siljković, Nebojša Denić, Dalibor Petković, Vuk Vujović, Natural Lignite Resources in Kosovo and Metohija and Their Influence on the Environment, Editor(s): Saleem Hashmi, Imtiaz Ahmed Choudhury. Encyclopedia of Renewable and Sustainable Materials, Elsevier, 2020, Pages561-566,ISBN9780128131961,https://doi.org/10.1016/B978-0-12-803581-8.11591-7. (https://www.sciencedirect.com/science/article/pii/B9780128035818115917)   

Denić, Nebojša,  Petković, Dalibor,  Siljković, Boris and  Ivković, Ratko (2020). Opportunities for Digital Marketing in the Viticulture of Kosovo and Metohija. In: Hashmi, Saleem and Choudhury, Imtiaz Ahmed (eds.). Encyclopedia of Renewable and Sustainable Materials, vol. 1, pp. 600–615. Oxford:Elsevier.http://dx.doi.org/10.1016/B978-0-12-803581-8.11592-9     

https://doi.org/10.1016/j.energy.2021.120621
https://doi.org/10.1016/j.compeleceng.2021.107270
https://doi.org/10.1016/j.rhisph.2021.100358
https://doi.org/10.1016/j.techfore.2021.120618
https://doi.org/10.1007/s13399-021-01314-2
https://doi.org/10.1007/s13399-020-01223-w
https://doi.org/10.1007/s13399-020-01014-3

Support vector regression methodology for wind turbine reaction torque prediction with power-split hydrostatic continuous variable transmission, Energy, Volume 67, April 2014, pp. 623–630;

Evaluation of modulation transfer function of optical lens system by support vector regression methodologies – A comparative study, Infrared Physics & Technology, DOI: 10.1016/j.infrared.2014.04.005, Volume 65, July 2014, pp. 94–102;

Sensor data fusion by support vector regression methodology – a comparative study, IEEE Sensors Journal, DOI: 10.1109/JSEN.2014.2356501, Volume 15, Issue 2, February 2015, pp. 850-854;

Forecasting of Underactuated Robotic Finger Contact Forces by Support Vector Regression Methodology, International Journal of Pattern Recognition and Artificial Intelligence, DOI: 10.1142/S0218001416590199, Volume 30, Issue 07, 2016, pp. 1–11;

ANALYZING OF CASE FATALITY RATE FORECASTING BY SOFT COMPUTING TECHNIQUE, ANNALS OF THE UNIVERSITY OF ORADEA, Fascicle of Management and Technological Engineering, Issue 3, December 2017, ISSN 2501-5796 (CD edition), ISSN 1583-0691 (online), ISSN-L 1583-0691 (online),)pp. 38-42.

Statistical evaluation of mathematics lecture performances by soft computing approach, Computer Applications in Engineering Education, DOI: 10.1002/cae.21931, Volume 26,  23 March 2018, pp. 902–905;

Analyzing of flexible gripper by computational intelligence approach, Mechatronics, DOI: 10.1016/j.mechatronics.2016.09.001, Volume 40, December 2016, pp. 1–16;

Selection of the most influential factors on the water-jet assisted underwater laser process by adaptive neuro-fuzzy technique, Infrared Physics & Technology, DOI: 10.1016/j.infrared.2016.05.021, Volume 77, July 2016, pp. 45–50;

Vibration analyzing in horizontal pumping aggregate by soft computing, Measurement, DOI: 10.1016/j.measurement.2018.04.100, Volume 125, September 2018,  pp. 454–462;

Prediction of laser welding quality by computational intelligence approaches, Optik - International Journal for Light and Electron Optics, DOI: 10.1007/s11760-016-0948-8, Volume 140, July 2017, pp. 597–600;

Precipitation concentration index management by adaptive neuro-fuzzy methodology, Climatic Change, DOI: 10.1007/s10584-017-1907-2, Volume 141, Issue 4, April 2017, pp. 655–669;

Estimation of fractal representation of wind speed fluctuation by artificial neural network with different training algorothms, Flow Measurement and Instrumentation, DOI: 10.1016/j.flowmeasinst.2017.01.007, Volume 54,  April 2017, pp. 172–176;

Adaptive neuro-fuzzy approach for wind turbine power coefficient estimation, Renewable and Sustainable Energy Reviews, Volume 28, December 2013, pp. 191-195;

Adaptive neuro-fuzzy maximal power extraction of wind turbine with continuously variable transmission, Energy, Volume 64, January 2014, pp. 868-874;

Adapting project management method and ANFIS strategy for variables selection and analyzing wind turbine wake effect, Natural Hazards, DOI: 10.1007/s11069-014-1189-1, Volume 74, Issue 2, November 2014, pp. 463-475;

Estimation of fractal representation of wind speed fluctuation by artificial neural network with different training algorothms, Flow Measurement and Instrumentation, DOI: 10.1016/j.flowmeasinst.2017.01.007, Volume 54,  April 2017, pp. 172–176;

Wind speed parameters sensitivity analysis based on fractals and neuro-fuzzy selection technique, Knowledge and Information Systems, DOI: 10.1007/s10115-016-1006-0

Wind farm efficiency by adaptive neuro-fuzzy strategy, International Journal of Electrical Power & Energy Systems, DOI: 10.1016/j.ijepes.2016.02.020, Volume 81, October 2016, pp. 215–221;

The research conducted here is not motivated.

Author Response

Dear Reviewer,

Thanks for your careful analysis of our work and relevant contributions. Please see the attachment.

Reviewer 2 Report

The topic is suitable for the journal. Still, the reviewer doubts that this paper is innovative enough to be published. The author should explain more about the main contribution of the manuscript. However, the following significant Major corrections seem necessary to improve the scientific level of the manuscript.

1- Please improve the abstract section; it doesn’t deliver enough information about the manuscript to readers. It should have a conclusion section.

2- Please explain more clearly the main contribution. The authors need to highlight the novelty of the work presented.

3- The introduction should be improved significantly. The introduction section format is not acceptable; the authors should re-write this section. The motivation of the manuscript should be highlighted in the introduction section.

4- Why do authors think they are appropriate for such an application? What is their main advantage?

5- The manuscript doesn’t have a good discussion or the conclusion section.

6- Please explain clearly the limitation and future work.  

7- The complexity of the proposed model and the model parameters uncertainty are not mentioned.

8-  A reasonable justification should be made about why such algorithms are used. Why do authors think they are appropriate for such an application? What is their main advantage over other methods? The literature review is not extensive. The literature review is long, but it is not content with important subjects. A detailed comparison with the performance of the work reviewed in the literature section might help understand the importance of the work presented.

Author Response

(The authors gave the same response as above.)

Reviewer 3 Report

1: Abstract: RL-SSI should be expanded when using for the first time.

     Why the JIGSAWS and Breakfast datasets are used for the study and significance.

     How the model can be generalized to different datasets?

2: Introduction: Showcase the additional research activities in the introduction, details should be mentioned on the contributions to the larger research community.

3: Materials and Methods Section: Explain the details on the implementation and explanation of RL-SSI is required in detail.

4: Explain the RL-SSI architecture and how it’s different from other already implemented algorithms / methods.

5: Datasets used in this research is used in any other research? And how are the results, just wanted to know the quality of the dataset to evaluate the methodology followed.

Author Response

(The authors gave the same response as above.)

Reviewer 4 Report

This paper presents an RL-SSI model: adapting a supervised learning approach to a semi-supervised method for human action recognition. The study primarily to mitigate the dependency of the RL-SSI model on annotated data and the iterative process used in reinforcement learning to recognize human activities in videos.

Provides a clear and descriptive statement of research design, sample, instruments, analysis, and procedures. 

The methodology/methods used provide adequate details systematically. 

Adequately identifies key experimental/computational parameters. Enough gives the level of accuracy, precision etc.

Author Response

(The authors gave the same response as above.)

Reviewer 5 Report

The paper has some oversights: 

The contribution of the paper should be outlined.
Reinforcement learning - semi-supervised and iterative learning should be described in the paper to guide the reader on this topic.
There is a lack of demonstration of the reward function used in the paper - please indicate this function.
Why was it decided to move only as many as 25% of the best labels to the training section? Please justify choice
What do short step ks and long step kl mean? - please explain this in the text of the paper
Table 1 is not adequately captioned - please indicate what is in it  

Author Response

(The authors gave the same response as above.)

Reviewer 6 Report

This paper is well-written. 

  • It is interesting to investigate the effect of not 100% label data. For example,80%,60%,40%,20%, and 0%.
  •  Please provide some insights into the obtained gain of the proposed method.
  • Two important paper on human motion recognition is missing, i.e., 1) Wireless Sensing With Deep Spectrogram Network and Primitive Based Autoregressive Hybrid Channel Model 2) Accelerating Edge Intelligence via Integrated Sensing and Communication. Please properly comment on these two papers in this paper. 

Author Response

(The authors gave the same response as above.)

Round 2

Reviewer 1 Report

Accept

Author Response

Thanks for your valuable contributions.

Reviewer 2 Report

The manuscript was modified very well. The authors have attempted to address all of the reviewers' comments in the revised paper. 

Author Response

Thanks for your valuable contributions.

Reviewer 5 Report

The authors conducted extensive revisions on their work. Therefore, I propose to accept the paper in its current state.

Author Response

Thanks for your valuable contributions.

Reviewer 6 Report

This paper needs to be strengthened, as many places are still imperfect.

1) The contributions of this paper are not clearly stated by points in the introduction.

2) “1- It is interesting to investigate the effect of not 100% label data. For example,80%,60%,40%,20%, and 0%. ” I do not agree with the authors that this is a future work. I suggest discuss one of them in this paper.

3) The references is not comprehence. Please comment the following paper:

  1. Wireless Sensing With Deep Spectrogram Network and Primitive Based Autoregressive Hybrid Channel Model
  2. Accelerating Edge Intelligence via Integrated Sensing and Communication. 

Otherwise, I do think this paper has sufficient merit for publication.

Author Response

Dear Reviewer,

Thanks for your valuable contributions. Please see the attachment. 

Round 3

Reviewer 6 Report

The author's reply well addresses my concerns. No further comments.